# Seq vs Seq: An Open Suite of Paired Encoders and Decoders

**Orion Weller**[ι]  **Kathryn Ricci**[ι]  **Marc Marone**[ι]
**Antoine Chaffin**[α]  **Dawn Lawrie**[ι]  **Benjamin Van Durme**[ι]

[ι] Johns Hopkins University    [α] LightOn

## Abstract

The large language model (LLM) community focuses almost exclusively on decoder-only language models, since they are easier to use for text generation. However, a large subset of the community still uses encoder-only models for tasks such as classification or retrieval. Previous work has attempted to compare these architectures, but is forced to make comparisons with models that have different numbers of parameters, training techniques, and datasets. We introduce the SOTA open-data ETTIN[1] suite of models: paired encoder-only and decoder-only models ranging from 17 million parameters to 1 billion, trained on up to 2 trillion tokens. Using the same recipe for both encoder-only and decoder-only models produces SOTA recipes in both categories for their respective sizes, beating ModernBERT as an encoder and Llama 3.2 and SmolLM2 as decoders. Like previous work, we find that encoder-only models excel at classification and retrieval tasks while decoders excel at generative tasks. However, we show that adapting a decoder model to encoder tasks (and vice versa) through continued training is subpar compared to using only the reverse objective (i.e. a 400M encoder outperforms a 1B decoder on MNLI, and vice versa for generative tasks). We open-source all artifacts of this study including training data, training order segmented by checkpoint, and 200+ checkpoints to allow future work to analyze or extend all aspects of training.[2]

## 1 Introduction

The rise of neural language models (LMs) was spurred by encoder-only models such as ELMo (Peters et al., 2018) and BERT (Devlin et al., 2019). However, the community generally shifted to decoder-only (i.e. GPT-style, ala Brown et al. (2020)) models due to their exceptional performance at sequence generation. Due to this lack of popularity for encoder-only models there was limited new model development, thus, we still frequently see usage of older models (i.e. from 2019) by the subset of the community focused on retrieval/classification or fast on-device inference. Although nascent work is attempting to revive encoder-only development (Samuel, 2024; Warner et al., 2024; Lee et al., 2025), there still exists a wide gap between the development of encoder-only and decoder-only models (synonymously referred to in this work as *encoders* or *decoders*).

Part of this gap is due to the sentiment within the community that decoders can be adapted for use in tasks that were once predominantly encoder-focused (e.g. classification, embeddings), especially as they can often be used in a zero-shot fashion (i.e. without fine-tuning) (BehnamGhader et al., 2024). As decoder models are more studied, more over-trained (Hoffmann et al., 2022), and are generally larger, they are now claiming the top spots of leaderboards for previously encoder-centric tasks (Enevoldsen et al., 2025).

Many works have challenged this assumption by comparing encoder-only and decoder-only models of roughly the same sizes (Ethayarajh, 2019; Charpentier & Samuel, 2024; Harrag et al., 2021). However, these analyses have to be done with incomparable models: using different architectures, different pre-training data, different learning schedules, etc.

---

[1] Named for the two-headed mythological Norse giant, symbolizing the two language models heads.
[2] Models, code, and data are available at `https://github.com/JHU-CLSP/ettin-encoder-vs-decoder`

Our work aims to provide the foundation to compare encoder-only and decoder-only models by open-sourcing a suite of models trained with the same data, the same architecture, and the same training recipe. Our ETTIN suite contains 10 models (5 pairs) ranging from 17 million to 1 billion parameters, and trained for up to 2 trillion tokens. This allows us to quantify the differences between these models (including the effects of scaling parameter size) in an apples-to-apples comparison.

Our models provide state-of-the-art performance for their size among open-data models. Surprisingly, they do so in both encoder settings (w.r.t. ModernBERT) and decoder settings (w.r.t. LLaMA 3.2 and SmolLM2) despite using the same recipe. Notably, our work also provides the first open-data replication of ModernBERT, allowing the community to further build upon our recipe.

We find that, like previous work, encoders excel at classification and retrieval while decoders excel at generative tasks. However, we go beyond previous work to examine the increasingly common setting (BehnamGhader et al., 2024) where decoder-models are continued trained as encoders (i.e. cross-objective training). We show results for this cross-objective training in both directions: training encoders for causal language modeling (CLM) and decoders with masked language modeling (MLM). We find that despite continued training for much longer than previous work (50B tokens) these models do not surpass those that started with this objective, i.e. a 400M encoder outperforms a 1B decoder continue-trained with MLM on MNLI, and vice versa for generative tasks.

Our work also provides the ability to compare these training objectives on other aspects, comparing how they learn. We provide a case study showing the effects of these objectives on gender bias.

Overall, our work provides the first suite of models enabling a fair comparison between encoder-only and decoder-only architectures (while also showing SOTA performance), enabling future work to analyze the effects of these training objectives on downstream tasks.

## 2 RELATED WORK

We describe encoder models as the community is generally more familiar with decoder LMs.[3] It is worth noting that our approach was inspired by Pythia (Biderman et al., 2023b) which was the first to explore open-data decoder-only models at multiple sizes.

**Encoder-only Models**   Encoder-only architectures were the predominant architecture for early transformer models, popularized by models such as BERT (Devlin et al., 2019), RoBERTa (Liu et al., 2019), and DeBERTa (He et al., 2023). These models showed significantly improved performance over the previous SOTA LSTM models on classification and retrieval tasks. This created a flurry of activity in the encoder space, with models improving on the BERT recipe: the RTD objective from DeBERTa, better data and objectives from RoBERTa, and many smaller variants such as TinyBERT (Jiao et al., 2019), DistilBERT (Sanh et al., 2019), BERT-small (Turc et al., 2019), and MiniLM L12 (Wang et al., 2020). However, these encoders lacked the easy ability to generate text and have generally fallen out of popularity in favor of decoder-only GPT-2 style models.

Despite this shift, encoders still maintain frequent usage for many tasks that don't require generative output. For example, in March 2025 alone, BERT-base had 90 million downloads on HuggingFace.

Recently, there has been renewed interest in encoders, as demonstrated by NomicBERT (Nussbaum et al., 2024), mosaicBERT (Portes et al., 2023), and ModernBERT (Warner et al., 2024). Unfortunately, ModernBERT (the most performant) does not provide access to their training data. Hence, we use publicly available data sources in order to replicate the training process.

**Comparisons between Encoders and Decoders**   Previous work has compared encoder-only and decoder-only models on a wide assortment of tasks. For example Charpentier & Samuel (2024) compare DeBERTa and GPT-2 in similar sizes. Other work (Yang et al., 2023; Qu et al., 2020; Zheng et al., 2021; Rehana et al., 2023; Nielsen et al., 2024) compares them on downstream tasks.

However, all of these comparisons have the same underlying limitation: the models they are comparing have different numbers of parameters, different architectures, different training recipes, and different

---

[3]For those interested in decoders, please see early works such as GPT-2 (Radford et al., 2019) and modern models such as OpenAI's GPT-4 (Achiam et al., 2023), Google's Gemini (Team et al., 2023), Alibaba's Qwen series Yang et al. (2025), and Meta's LLaMA models (Grattafiori et al., 2024)

| Parameter | 17M (XXS) | 32M (XS) | 68M (Small) | 150M (Base) | 400M (Large) | 1B (XL) |
|---|---|---|---|---|---|---|
| Layers | 7 | 10 | 19 | 22 | 28 | 28 |
| Hidden Size | 256 | 384 | 512 | 768 | 1024 | 1792 |
| Intermediate Size | 384 | 576 | 768 | 1152 | 2624 | 3840 |
| Attention Heads | 4 | 6 | 8 | 12 | 16 | 28 |
| Learning Rate | 3e-3 | 3e-3 | 3e-3 | 8e-4 | 5e-4 | 5e-4 |
| Weight Decay | 3e-4 | 3e-4 | 3e-4 | 1e-5 | 1e-5 | 5e-5 |
| Warmup Tokens (B) | 4 | 4 | 3 | 3 | 2 | 2 |
| BS Warmup (B) | 125 | 100 | 75 | 50 | 10 | 3 |

Table 1: Configuration for each Ettin model size. Both encoders and decoders use the exact same configuration, differing only in attention (bidirectional vs causal) and objective (MLM vs CLM).

pre-training data. Although some work has attempted to address this (Charpentier & Samuel, 2024; Gisserot-Boukhlef et al., 2025), they have only done so in limited settings with very small amounts of pre-training data. In contrast, we train SOTA models, allowing for an exact comparison.

**Bidirectional Attention for Decoders**   Although we cannot cover it all here, there have been attempts to use bidirectional attention for standard decoder usage. This includes prefix LM attention (Artetxe et al., 2022; Chowdhery et al., 2023; Du et al., 2021) and other mixed training such as BiTune (Kopiczko et al., 2024). However, most modern LM training still favors pure CLM.[4]

**Checkpoint-Level Model Analyses**   There has also been much work exploring how models learn via their training data. This was popularized by the Pythia (Biderman et al., 2023a) paper and includes many aspects of learning such as data quality and selection (Longpre et al., 2024), how frequency of entities impacts model learning (Oh et al., 2024), effects of the recency of data (Cheng et al., 2024), and whether you can recognize and extract training data from models (Zhang et al., 2024). Our work allows these experiments to be done on more recent SOTA models and provides a way to compare encoders and decoders on various facets of learning.

## 3   EXPERIMENTAL SETTINGS

### 3.1   TRAINING DATA

We create an open-source replication of ModernBERT (Warner et al., 2024) due to it being the strongest publicly available encoder-only model and using comparable techniques to decoder-only models. This provides the best starting place for a recipe that spans both training objectives. However, ModernBERT's data is not publicly available – thus, we aim to replicate the recipe using open-data.

We do so by pulling from the best publicly available datasets used for training decoder-only models, such as Olmo (Groeneveld et al., 2024; OLMo et al., 2025). Thus, we use a mix of DCLM (Li et al., 2024) combined with various curated sources from Dolma v1.7 (Soldaini et al., 2024). In the process of training, the Olmo 2 paper (OLMo et al., 2025) described their approach of using filtered DCLM and other higher-quality sources for the decay phase (similar to FineWeb-Edu filtering Penedo et al. (2024); Lozhkov et al. (2024)). We decided to use these newer sources for our later phases.[5]

To allow others to easily extend our work, we provide both formats: the data which can be used for training as well as the data seen by the models in batch order for future analyses.

### 3.2   ARCHITECTURE

As ModernBERT has only two sizes, we develop new shapes for our smaller and larger models (Table 1). We aim to follow the design espoused by MobileLLM (Liu et al., 2024) with deep but thin models. However, for the 1B model, we keep the same number of layers but make the model wider.

---

[4]To the best of our knowledge, as much of the details of the best LMs now goes unpublished.

[5]We ablated with the non-filtered data and found worse results.

| | | Pre-training | | Mid-training | | Decay Phase | |
|---|---|---|---|---|---|---|---|
| Category | Dataset | Tokens (B) | % | Tokens (B) | % | Tokens (B) | % |
| News | CC News | 7.3 | 0.4 | – | – | – | – |
| Code | Starcoder | 263.9 | 15.5 | 38.4 | 15.4 | – | – |
| Code | Code_Repos | – | – | – | – | 20.2 | 26.5 |
| Crawl | CC Head | 356.6 | 20.9 | – | – | – | – |
| Crawl | DCLM | 837.2 | 49.1 | – | – | – | – |
| Crawl | DCLM (Dolmino) | – | – | 175.5 | 70.4 | 26.0 | 34.1 |
| Math | Open-Web-Math | 12.7 | 0.7 | – | – | – | – |
| Math | Algebraic StackExchange | 12.6 | 0.7 | – | – | – | – |
| Math | Math (Dolmino) | – | – | 10.4 | 4.2 | 5.0 | 6.6 |
| Scientific | PeS2o | 57.3 | 3.4 | 8.3 | 3.3 | – | – |
| Scientific | Arxiv | 28.0 | 1.6 | 4.1 | 1.6 | 3.0 | 3.9 |
| Social | Reddit | 80.3 | 4.7 | 6.2 | 2.5 | – | – |
| Social | StackExchange | 19.6 | 1.1 | – | – | – | – |
| Social | StackExchange (Dolmino) | – | – | 2.7 | 1.1 | 4.0 | 5.2 |
| Reference | Textbooks | – | – | – | – | 0.5 | 0.7 |
| Reference | Dolma Books | 5.3 | 0.3 | 0.8 | 0.3 | 10.5 | 13.8 |
| Reference | Wikipedia | 7.3 | 0.4 | 0.5 | 0.2 | 3.0 | 3.9 |
| Instruction | Tulu Flan | 16.6 | 1.0 | 2.4 | 1.0 | 4.1 | 5.4 |
| **Total** | | 1,704.7 | 100.0 | 249.3 | 100.0 | 76.3 | 100.0 |

Table 2: Training data mixture across the various training stages (pre-training, mid-training, decay). Later stages use higher quality data, from the recently released Dolmino dataset (OLMo et al., 2025). Dashes indicate that no data from that source was used. We trained for 1.7T tokens for pre-training, 250B for mid-training, and 50B for the decay phase. We sample from the dataset and repeat (or under-sample) as needed to hit the token counts used for training.

We choose models parameter sizes at roughly 2x increments while matching common encoder sizes, e.g. 17M, 32M, 68M, 150M, 400M, and 1B. For a detailed list of the differences, see Table 1.

## 3.3 TRAINING RECIPE

We use the same general process described by open-data models (which was followed by Modern-BERT) for training both encoder-only and decoder-only models – with a few specific changes for the encoder architecture (i.e. masking ratio). In summary, we include three general phases: base pre-training, mid-training/context extension, and decay. See Table 2 for the precise sources of training data in each phase. We use a trapezoidal learning rate scheduler, with general hyperparameters shown in Appendix C and size-dependent hyperparameters in Table 1. For compute details see Appendix E.

The only differences between the encoder and decoder models are: (1) the objective function, i.e. masked language modeling (MLM) for the encoder[6] vs causal language modeling (CLM) for the decoder and (2) the attention pattern, i.e. causal for the decoder and bidirectional for the encoder.

We checkpoint every 8.5B tokens, with 236 checkpoints per model. Combined with the batch ordering of the data, this enables precise pinpointing of what the model learned between each checkpoint.

**Base Pre-training** This stage encompasses the warmup and stable phase of the trapezoidal learning rate, training for 1.7T tokens. We use both learning rate and batch size warmup. The data in this stage comprises a wide mix of sources to allow for general learning.

**Context Extension / Mid-Training** In this phase we increase the quality of the data and change both the data and base RoPE (Su et al., 2024) to handle longer context. We update the data length to be up to 8000 tokens and RoPE parameters to 160k (for global and local layers). For the data, we drop the noisiest sections (older Dolma common crawl, CC News, general StackExchange) and include filtered DCLM, math, and StackExchange. We then train for 250B tokens and use an inverse square root learning rate schedule from the peak learning rate to 1/2 of the peak.

---

[6]For the encoder we use a 30% masking ratio except for the decay phase, which is 15%.

**Decay Phase**    Finally, we use one more inverse square root learning rate schedule to decay for 50B tokens. We follow the general ProLong recipe (Gao et al., 2024b), increasing long context data such as Dolma books, Wikipedia, and open-access textbooks. We decay to 0.02 of the peak LR.

## 3.4    Major training differences from ModernBERT

A concise summary of the largest differences from the ModernBERT recipe are (1) the use of open-data, (2) decay in the context extension phase, (3) no model merging,[7] (4) a lower masking ratio for the decay phase (15% instead of 30%) and (5) local and global RoPE to be the same value.

## 3.5    Cross-Objective Training

As encoder models have gone out of popularity, decoder models have increased in size (both parameters and pre-training data). Thus, these newer decoder models are typically trained for much longer than previous encoder models (i.e. BERT). Due to this it has become common to adapt these larger decoder models to what were previously encoder-centric tasks (Zhang et al., 2025). With paired encoder and decoder models, we can now answer the question of **how effective this continued pre-training approach is** and **whether it is still worth training both types of models**. We call this *cross-objective training*: taking the final model and continue pre-training it on the reverse objective. Following BehnamGhader et al. (2024), we do not use MLM but rather use MNTP, that is, the masked token is predicted using the hidden state of the previous token to better align with CLM.

We train for 50B tokens with the reverse objective, which is far more than previous work has attempted e.g. LLM2Vec (BehnamGhader et al., 2024), which is around 10B tokens. Although the ratio of pre-training and cross-objective training is unbalanced, this mimics the realistic setting where the adaptation is done with very small amounts of data comparatively. We do this cross-objective training on the highest quality data we have available, which was used in the last decay phase.[8] We use a new trapezoidal learning rate schedule with 3B tokens of warmup and 10B tokens of decay.[9] Thus, by the end we have an encoder-from-decoder (i.e. a decoder further pretrained with MNTP similar to LLM2Vec[10]) and a decoder-from-encoder (i.e. an encoder further pre-trained with CLM).

## 4    Experiments

We aim to compare encoder and decoder models. However, first, to give those experiments credence, we show that our models are SOTA. This strengthens our claim and helps alleviate concerns that we made training choices that favored one architecture over the other – instead we have SOTA models in both architectures for their sizes, showing our method's effectiveness. Note though, that the purpose of our paper was not to be SOTA overall (i.e. compared to OpenAI, etc.), but to provide a comparison for encoders and decoders. For space and to avoid repetition, specific model size details are in Table 1.

## 4.1    Individual Evaluations

**Encoder-Only Results**    We use two baselines for each size type: extra extra small (XXS) BERT-mini and TinyBERT, extra small (XS) models MiniLM L12 and BERT-small, small (S) models DistilBERT and DistilRoBERTa, base (B) models BERT and ModernBERT, large (L) models BERT-large and ModernBERT-large, and an extra large (XL) model DeBERTA v2 XL.[11]

We evaluate on various encoder tasks, including GLUE (Wang et al., 2018), MTEB v2 English (Enevoldsen et al., 2025), MDLR for long context (Chen et al., 2024), and CodeSearchNet for code

---

[7]We do this for ease of scientific comparison, however, if one was to use this for downstream applications a simple merge would likely boost performance another point or two.

[8]We note that this means it repeating this data twice, however, as shown by previous work (Muennighoff et al., 2023) two repetitions on high quality data has no adverse effects.

[9]For the 1B model this is scaled by 1/3 again due to compute availability.

[10]We use a 15% masking rate for the encoder-from-decoder as to maintain a middle ground masking ratio.

[11]We also ran experiments with DeBERTa XXL as shown in the appendix. However, due to the size and slowness of the architecture we could not do a comparable grid search. Our results in the appendix are after 300 days of H100s hours, but still did not complete the full sweep. Furthermore, as DeBERTa XXL is > 1.5B we exclude it as it is significantly larger than 1B (i.e. > 50% larger).

| Model Name | CSN | MLDR | Embedding Tasks Clustering | Retrieval | MTEB v2 | GLUE Tasks SST-2 | MNLI | GLUE Avg |
|---|---|---|---|---|---|---|---|---|
| **XXS Models (7-17M parameters)** | | | | | | | | |
| BERT-mini | 41.3 | 16.8 | 39.0 | 34.7 | 49.2 | 88.3 | 77.2 | 76.4 |
| TinyBERT | 39.8 | 14.2 | 37.4 | 33.3 | **49.7** | **91.2** | **80.9** | 77.0 |
| Ettin-Enc-17m | **59.1** | **24.4** | **39.1** | **35.6** | 48.9 | **91.2** | 79.5 | **79.2** |
| **XS Models (28-33M parameters)** | | | | | | | | |
| BERT-small | 46.0 | 19.9 | **39.6** | 38.1 | 51.1 | 90.1 | 79.2 | 79.0 |
| MiniLM L12 | 48.3 | 19.6 | 37.8 | 38.4 | **51.3** | **93.3** | **85.6** | **84.6** |
| Ettin-Enc-32m | **69.2** | **28.4** | **39.6** | **39.7** | 50.9 | 92.0 | 83.4 | 83.5 |
| **Small Models (68-82M parameters)** | | | | | | | | |
| DistilBERT | 47.9 | 23.7 | 39.8 | 40.8 | **52.7** | 92.2 | 82.7 | 81.5 |
| DistilRoBERTa | 60.3 | 19.7 | 39.3 | 40.0 | 51.8 | 93.1 | 84.7 | 83.8 |
| Ettin-Enc-68m | **75.1** | **30.1** | **40.1** | **43.1** | 52.6 | **94.4** | **87.0** | **87.2** |
| **Base Models (123-150M parameters)** | | | | | | | | |
| BERT base | 51.0 | 24.8 | 40.4 | 41.2 | 52.9 | 93.1 | 85.4 | 84.7 |
| ModernBERT base | 75.9 | 30.4 | 41.3 | 43.9 | **54.0** | **96.0** | 89.1 | 88.4 |
| Ettin-Enc-150m | **76.3** | **31.8** | **41.5** | **45.7** | **54.0** | 95.8 | **89.2** | **88.9** |
| **Large Models (353-395M parameters)** | | | | | | | | |
| BERT large | 54.4 | 25.3 | 41.5 | 42.9 | 53.8 | 93.3 | 86.3 | 85.2 |
| ModernBERT large | 78.3 | 34.9 | 41.5 | 47.0 | 55.0 | **97.1** | 90.8 | 90.4 |
| Ettin-Enc-400m | **80.7** | **36.2** | **41.8** | **48.4** | **55.5** | 96.7 | **91.3** | **90.8** |
| **XL Models (884M-1.2B parameters)** | | | | | | | | |
| DeBERTa-v1-xl | 75.6 | 28.1 | **42.5** | 47.2 | **56.4** | **97.1** | 91.7 | 90.7 |
| Ettin-Enc-1B | **82.3** | **40.2** | 41.9 | **50.1** | 56.0 | **97.1** | **91.8** | **91.6** |

Table 3: ETTIN encoders compared to other encoder-only models across various sizes on retrieval and GLUE tasks. Due to space, we show two representative tasks from MTEB v2 and two from GLUE, as well as a code-based retrieval evaluation (CodeSearchNet) and a long-context evaluation (MLDR). See Appendix A for the full tables of GLUE and MTEB v2. **ETTIN shows significant gains over baseline encoders, including ModernBERT, while also having both larger and smaller sizes.**

evaluation (Husain et al., 2019). We use the same evaluation setup as ModernBERT for the evaluation for an equal comparison (see Appendix F for hyperparameter details).

We find in Table 3 that ETTIN compares favorably overall. The relatively larger gains in the bigger sizes is likely because the smaller model baselines are heavily optimized with distillation.[12] Even so, we see that they generally outperform the baselines without doing any distillation: e.g. Ettin-68m with a GLUE average of 87.2 compared to the next best DistilRoBERTa at 83.8. Even for the more recent ModernBERT baselines we see improved performance (88.9 GLUE average vs 88.4 for the base size). Thus we can see that ETTIN matches or improves the SOTA for encoder-only models.

**Decoder-Only Results** We use two baselines for each size type when available, but few very small decoder-only LMs exist: One extra extra small (XXS) model Pythia-14M,[13] no models in the extra small (XS) category that we could find, one small (S) model DistilGPT (Sanh et al., 2019), two base (B) models Pythia 160M (Biderman et al., 2023a) and SmolLM2 135M, two large (L) models Pythia 410M and SmolLM2 360M (Allal et al., 2025), and extra large (XL) models Olmo 1B 0724 Groeneveld et al. (2024) and Llama 3.2 1B (Dubey et al., 2024).[14]

We evaluate on a wide range of tasks using the Eleuther AI harness (Gao et al., 2024a) (see Appendix B for details), consolidating tasks used in the Pythia and SmolLM papers including: the ARC Challenge (ARC) Clark et al. (2018), HellaSwag (HS) (Zellers et al., 2019), LAMBADA (LMB) (Paperno et al., 2016), OpenBookQA (OBQA) (Mihaylov et al., 2018), Social IQA (SIQA) (Sap et al., 2019), TriviaQA (TQA) (Joshi et al., 2017), Winogrande (WG) (Sakaguchi et al., 2021), and the

---

[12]We also note that MiniLM L12 has twice the amount of non-embedding parameters 21M vs 12M.

[13]This is likely just a debug-sized run and not an official size, as they do no include it in their paper. However, as not other decoder models in this size could be found, we use it as a reference.

[14]What is considered "1B" has a large range, up to nearly 2B parameters. We thus restrict our range to < 1.2B parameters to be a "1B" model, which excludes models like SmolLM2 1.7B and Olmo 2 1B (actually 1.5B).

| Model Name | ARC | HS | LMB | OBQA | PIQA | SciQ | SIQA | TQA | WG | WSC | Avg |
|---|---|---|---|---|---|---|---|---|---|---|---|
| **XXS Models (14-17M parameters)** | | | | | | | | | | | |
| Pythia-14m | 21.2 | 26.0 | 7.1 | 26.2 | 55.2 | 43.8 | 33.4 | 0.0 | 50.3 | **51.6** | 31.5 |
| Ettin-Dec-17m | **21.3** | **27.1** | **23.0** | **27.2** | **57.7** | **71.1** | **35.4** | **2.6** | **50.9** | 48.0 | **36.4** |
| **XS Models (32M parameters)** | | | | | | | | | | | |
| Ettin-Dec-32m | **23.5** | **28.5** | **28.5** | **28.2** | **57.7** | **77.5** | **36.4** | **3.8** | **53.1** | **50.2** | **38.7** |
| **Small Models (68-82M parameters)** | | | | | | | | | | | |
| DistilGPT | 23.0 | 27.5 | 25.0 | 26.8 | 59.8 | 62.6 | 36.1 | 0.3 | **50.4** | 53.8 | 36.5 |
| Ettin-Dec-68m | **25.3** | **33.4** | **35.2** | **29.4** | **61.8** | **83.2** | **38.8** | **5.6** | 50.1 | **55.3** | **41.8** |
| **Base Models (135-160M parameters)** | | | | | | | | | | | |
| SmolLM2-135m | **29.1** | **43.1** | 42.9 | **32.4** | **68.4** | 78.5 | 39.4 | 5.0 | **53.7** | **59.7** | 45.2 |
| Pythia-160m | 24.0 | 30.2 | 32.9 | 26.4 | 62.0 | 67.2 | 36.9 | 0.4 | 52.4 | 58.2 | 39.1 |
| Ettin-Dec-150m | 28.6 | 40.3 | **43.2** | 29.2 | 66.6 | **89.6** | **40.1** | **11.2** | **53.7** | 59.0 | **46.2** |
| **Large Models (360-410M parameters)** | | | | | | | | | | | |
| Pythia-410m | 24.7 | 40.6 | 51.5 | 29.4 | 67.0 | 72.3 | 39.0 | 1.8 | 53.6 | 65.2 | 44.5 |
| SmolLM2-360m | **37.6** | **56.3** | **53.5** | **37.6** | **71.8** | 86.6 | 40.7 | **18.4** | **58.6** | 70.3 | **53.1** |
| Ettin-Dec-400m | 33.6 | 54.3 | 52.3 | 34.4 | 71.0 | **91.8** | **45.5** | 18.3 | 57.6 | **71.8** | **53.1** |
| **XL Models (908M-1.2B parameters)** | | | | | | | | | | | |
| OLMo-1B-0724 | 32.3 | **66.1** | 61.0 | 35.6 | **75.1** | 91.8 | **49.2** | 1.2 | 61.6 | 76.9 | 55.1 |
| Llama-3.2-1B | 36.2 | 63.7 | **62.1** | 37.2 | 75.0 | 88.4 | 43.2 | 24.9 | 60.6 | 74.7 | 56.6 |
| Ettin-Dec-1B | **39.7** | 62.9 | 58.4 | **41.6** | 74.4 | **93.8** | 48.2 | **29.3** | **62.7** | **79.1** | **59.0** |

Table 4: Performance comparison of **decoder-only** models across tasks, organized by size categories. **We see that ETTIN decoders compare favorably, matching or exceeding the previous open-data SOTA**. Task names in order are ARC, Hellaswag, LAMBADA, OpenBookQA, Social IQA, TriviaQA, Winogrande, and Winograd Schema Challenge.

Winograd Schema Challenge (WSC) (Levesque et al., 2012). Note all tasks use zero-shot closed-book evaluations.

We see the results in Table 4 and see that ETTIN performs well compared to baseline models, such as ETTIN-150m outperforming SmolLM2 46.2 to 45.2 and ETTIN-1B's 59.0 to Llama 3.2 1B's 56.6 average). Thus we can see that ETTIN improves the SOTA for open-data decoder-only models.

## 4.2 ENCODERS VS DECODERS

Now that the strength of the training recipe is established, we can compare the two training objectives.

For simplicity, we show the two most representative encoder tasks (MS MARCO dev Bajaj et al. (2016) for retrieval and classification on MNLI) and keep the average generative score.[15] We evaluate the decoders on encoder-only tasks and vice versa, and similarly with the cross-objective trained models. We evaluate encoder-only models on decoder generative tasks using the method proposed in Samuel (2024), i.e. using three mask tokens at the end of the sequence and filling in the first token iteratively. Figure 1 shows the results of this comparison across models sizes.

**MNLI Classification** On the representative classification task, we see that encoders dominate decoders, as typically found. Furthermore, we see that even cross-objective continued pre-training does not resolve this gap, with enc-from-dec performance remaining similar to the original decoder model. Furthermore, the pure encoder models are typically better than the next larger sizes of decoders, e.g. the 150M encoder scoring 89.2 compared to the 400M decoder's 88.2.

**MS MARCO Dev Retrieval** For retrieval we see similar encoder dominance like classification, but notably improved performance when continue pre-training the decoder (i.e. the encoder-from-decoder). The MTNP continue pre-training significantly helps the decoder at all sizes, yet even the additional 50B tokens of pre-training is not enough to match the performance of the encoder (i.e. for the 400M size we have 42.2 for the encoder vs 41.4 for the encoder-from-decoder). Although the

---

[15]As decoder evaluations are significantly quicker than fine-tuning which is required for encoder tasks.

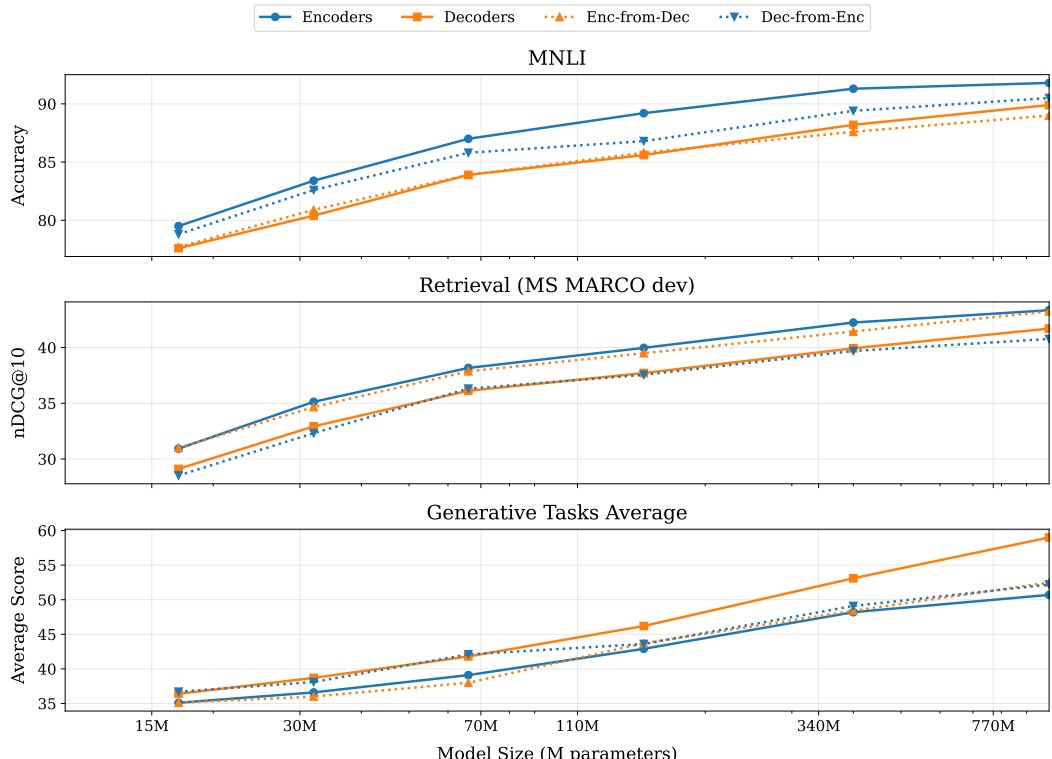

Figure 1: Encoder vs decoder comparison across model size. Generally **models in the preferred architecture (e.g. encoders in MNLI) do better than the opposite architecture even with an order of magnitude greater size**, e.g. a 400M decoder outperforming the 1B encoder. Notably, in generative tasks, decoders-from-encoders scale poorly with size.

difference is not as pronounced as in classification, we find that continued pre-training a decoder for retrieval is still subpar compared to simply using an encoder, even despite the additional 50B tokens.

**Generative Tasks** We find the reverse of the previous tasks: decoders do better than the decoder-from-encoder in general, with a widening gap (from similar scores at 68m parameters to greater than a 6 point difference at 1B) as model size increases. Notably, it appears that continued training of encoders-from-decoders scales poorly, perhaps why there is little-to-no previous work on the topic.

Despite this, we note that this average hides some nuance: on "generative" tasks that are more classification focused (such as ARC and SciQ) encoder models used in a generative fashion actually exceed decoder performance (i.e. for the 400M size the encoder scores 35.6 ARC vs the decoder's 33.6). However, decoders show huge gains on tasks such as HellaSwag, TriviaQA, and SiQA, making it so the average is strongly in favor of the decoders. See Table 8 for all sub-task results.

Although most of our models are too small to have signal on more difficult, but standard LM tasks like MMLU and GSM8k, we evaluate these for just the 1B sized models in Table 5. We find similar results to the previous table, illustrating that the Decoder-From-Encoder significantly outperforms on MMLU classification but is significantly worse on the generative GSM8k.

| Model | MMLU CS | GSM8k |
|---|---|---|
| Decoder | 27.0 | **32.0** |
| Dec-From-Enc | **37.0** | 18.9 |

Table 5: Harder benchmark results for 1B models on reasoning and classification tasks.

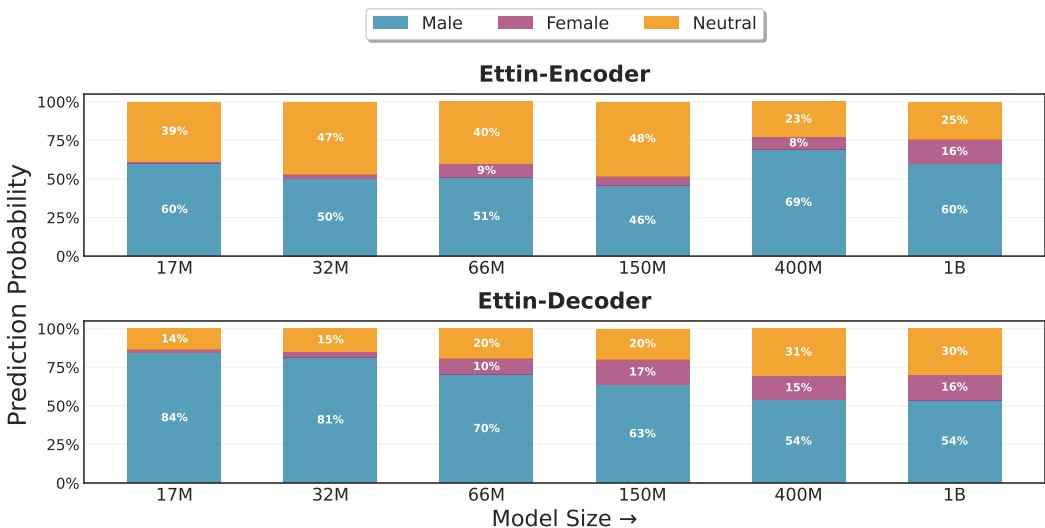

Figure 2: Gender pronoun predictions on the Gotcha split of WinoGender (Rudinger et al., 2018), a 50/50 stereotypical male/female split. **We see that encoder models are more likely to use gender neutral pronouns whereas both are biased towards male pronouns.**

## 5 CASE STUDY: GENDER BIAS

Due to the Ettin suite's open-pretraining data, we can also analyze other aspects of learning across pre-training objectives. As one example, we analyze gender representations for bias.

We use the WinoGender benchmark (Rudinger et al., 2018) using the "Gotcha" split that has a 50/50 split of male/female stereotypical pronouns (i.e. female for nurse). However, the standard coreference task is hard for most of our small models. Thus, we show results for an easier task: simply predicting the pronoun in the sentence. For the standard coreference task results, see Table G in the appendix.

We have each model predict the pronouns (i.e. by using a mask token for encoders or by choosing the lower perplexity sentence with decoders) and show the distribution of predicted pronouns per model (male, female, or gender neutral).[16] The results are in Figure 2, which shows that encoders are much more likely to use a gender neutral pronoun overall. In both encoders and decoder, female pronouns become more used as the size of the model gets larger: for decoders there is a clear trend of progressively smaller amounts of male pronouns, whereas for encoders the trend is more stochastic. For effects of the cross-training objectives on the model, see Figure 3 in the Appendix.

**Overall** As both models had the same training data, we find that the MLM objective leads the model to choose more neutral pronouns over female pronouns. However, male gender bias seems strong in both models, if slightly higher for decoders. Thus, this is one example of the analysis enabled by our data; we leave others to future work.

## 6 DISCUSSION

Our work suggests the following conclusions: (1) as speculated by previous work, MLM and CLM objectives do convey different strengths – MLM for classification and retrieval and CLM for generative tasks. However, (2) we also went a step further to show that simply continued pre-training on the reverse objective does not make up the difference from not using the preferred architecture.

This has several implications for those using models for classification or retrieval: currently the top models on leaderboards like MTEB are 7B+. However, based on our experiments, it is likely a 3B encoder model would outperform it. But, the lack of large encoder-only models means that approaches that continue pre-train decoders using MLM will likely outperform all other options (as is currently seen on the leaderboards). In the small scale regime (1B or less) where it is easier to

---

[16]There are more than three types of pronouns used in English beyond what is in this dataset. However, WinoGender is only designed for these three. We leave extensions of this dataset to future work.

train more "niche" encoder models for classification/retrieval, our results indicate that encoders will continue to outperform all others in their size range (and even ones above it).

Our results also suggest that encoders and decoders learn differently in other aspects as well, such as gender bias. Although this is just one example, we look forward to future research that discovers other differences. Overall, our artifacts allow for a range of new analyses and pre-training research.

We note that concurrent work (Gisserot-Boukhlef et al., 2025) did a similar analysis and found that starting from CLM and doing continued MLM pre-training was better in nearly all cases. However, pre-training was done with only 100B tokens; thus it is likely that this is an artifact of CLM's comparative data efficiency in the smaller data regime (i.e. loss on every token compared to loss on X% of tokens for MLM). Our work shows that at scale, for SOTA models, encoders do greatly outperform encoders-from-decoders (as also shown by SOTA concurrent work Marone et al. (2025)).

## 7 CONCLUSION

We provide the first suite of paired models that use the same training recipe to compare encoder-only and decoder-only models. Our models are SOTA in their size for open-data models, and are the first public recipe for ModernBERT-style models. We show that encoders are strong in classification and retrieval, while decoders are strong in generative tasks. Furthermore, we show that this difference can not easily be solved by continued training with the reverse objective. We show that this suite allows the analysis of how pre-training objective impacts learning, showing a case study in gender bias. We release all artifacts (including training data order) to help future researchers analyze these models.

## ACKNOWLEDGMENTS

This work has been supported by both DARPA SciFy and the U.S. National Science Foundation under grant 2204926. Any opinions, findings, and conclusions or recommendations expressed in this article are those of the authors and do not necessarily reflect the views of the National Science Foundation or DARPA. OW is supported by an NSF GRFP fellowship.

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

## A    ALL ENCODER RESULTS

We show results for the full MTEB v2 eng and GLUE results in Table 6 and Table 7 respectively.

## B    DECODER EVALUATION FRAMEWORKS

For generative tasks, we use the Eleuther AI harness with commit `867413f8677f00f6a817262727cbb041bf36192a`. We also use a forked version of the Eleuther AI harness for evaluating encoders in a generative fashion (see Github for details). Following previous work (Allal et al., 2025) the ARC score is the average of the easy and challenge sets.

For encoders evaluated on generative tasks we use three mask tokens followed by the EOS token. At each step, we predict the first MASK token and iteratively generate. However, this approach still could be improved, in particular around the EOS token. Encoder models are non-calibrated for when this should appear, so we had to make small changes to this setup for two tasks: for TriviaQA we change the EOS token to a newline character (as the harness stops on newlines for TriviaQA also) and for the Lambada OpenAI task we do not score the EOS token. All other tasks proceed with the three masks + EOS token as proposed by Samuel (2024).

## C ARCHITECTURE DETAILS

Architecture and training details for all models are found in Table G. These are generally the same as ModernBERT except for the same value of local and global RoPE and a slightly shorter context length (7999).

## D MODEL SIZES

Model sizes (both embedding and non-embedding) are found in Table 11. We group models by total parameters, although we note that some have more vocab parameters vs non-vocab parameters, e.g. MiniLM L12 has almost 2x the number of non-embedding parameters compared to Ettin-32m (21M vs 12M).

## E COMPUTE CONFIGURATION

We train the models on a comparatively small compute cluster. We train each model on a 4xH100 node using NVLink. The pre-training phase (the longest) takes approximately 6 days for the 17M model. The longest was for the 1B model, which we trained for approximately 40 days. Unfortunately, we did not have enough compute availability to train the 1B to the full 2T tokens. Thus the 1B models are scaled to 1/3 of the data (e.g. 667B instead of 2T tokens). However, this is still more than chinchilla optimal (Hoffmann et al., 2022) and it still outperformed other 1B models trained longer.

## F ENCODER EVALUATION SWEEP PARAMETERS

Below we detail the sweep hyperparameters for the retrieval and classification tasks that require fine-tuning.

**GLUE** We re-use ModernBERT's evaluation setup but slightly increase the learning rate sweep in order to better fit the smaller parameter models (which typically use higher LRs). As the best LRs chosen by BERT and ModernBERT were lower than these, it does not affect their scores. We sweep for learning rates over {1e-5, 3e-5, 5e-5, 8e-5, 1e-4}, weight decay values over {1e-6, 5e-6, 8e-6, 1e-5}, and batch sizes over {16, 32}. We also sweep over epochs {1, 2, 3, 4} if task $\in$ {mnli, sst2, rte}, otherwise {2, 5, 10, 12}. This was a total of 160 sweeps per model, of which we select the best score per task per model to report. We start from the best MNLI checkpoint for fine-tuning on RTE, STS-B, and MRPC, following ModernBERT.

**Retrieval** We sweep four LRs ({1e-4, 3e-4, 5e-4, 7e-4}) on MS MARCO dev and choose the best performing one to evaluate on the other retrieval datasets. We use a new retrieval training script due to being unable to exactly reproduce ModernBERT's precise scores. While doing so, we also improve the training process for all models due to the use of more negatives in training, achieving higher scores than that in the ModernBERT paper (which was not trying to optimize scores, but shows generally that our training script is effective). We trained with an effective batch size of 1024 with 4 accumulation steps. For DeBERTa-v2 it diverged for all learning rates we tried. Thus, to get it to converge, we changed the warmup to 20% from 5% and lowered the learning rate to 1e-5.

As Ettin models have been trained with instructions during pre-training, it is likely they are also more capable for instruction-based retrieval (Shao et al., 2025; Weller et al., 2024; 2025), however, we leave that for future work.

## G LLM USAGE

LLMs were not used for paper writing, but were used for coding assistance and title brainstorming. All code was human verified.

| Model Name | Mean (Task) | Mean (Type) | Class. | Clus. | Pair. Class. | Rerank. | Retrieval | STS | Summ. |
|---|---|---|---|---|---|---|---|---|---|
| **XXS Models (11-25M parameters)** | | | | | | | | | |
| TinyBERT | 52.1 | 49.7 | 64.6 | 37.4 | 78.1 | 41.9 | 33.3 | 71.9 | 20.6 |
| BERT-mini | 51.8 | 49.2 | 61.7 | 39.0 | 77.8 | 41.5 | 34.7 | 70.8 | 19.3 |
| Ettin-Enc-17m | 52.3 | 48.9 | 63.3 | 39.1 | 74.8 | 42.0 | 35.6 | 71.6 | 16.0 |
| **XS Models (28-33M parameters)** | | | | | | | | | |
| BERT-small | 54.0 | 51.1 | 64.7 | 39.6 | 79.4 | 41.7 | 38.1 | 73.0 | 21.6 |
| MiniLM L12 | 53.9 | 51.3 | 64.9 | 37.8 | 79.8 | 43.5 | 38.4 | 73.2 | 21.4 |
| Ettin-Enc-32m | 54.2 | 50.9 | 64.2 | 39.6 | 77.1 | 42.6 | 39.7 | 73.2 | 19.9 |
| **S Models (68-82M parameters)** | | | | | | | | | |
| DistilBERT | 55.5 | 52.7 | 66.5 | 39.8 | 80.7 | 42.8 | 40.8 | 74.1 | 24.0 |
| DistilRoBa | 55.2 | 51.8 | 67.3 | 39.3 | 78.9 | 43.5 | 40.0 | 74.2 | 19.4 |
| Ettin-Enc-68m | 56.1 | 52.6 | 66.6 | 40.1 | 79.3 | 43.3 | 43.1 | 74.2 | 21.7 |
| **Base Models (86-150M parameters)** | | | | | | | | | |
| BERT-base | 56.0 | 52.9 | 67.2 | 40.4 | 80.5 | 43.1 | 41.2 | 74.8 | 23.1 |
| ModernBERT-base | 57.1 | 54.0 | 67.5 | 41.3 | 80.4 | 44.7 | 43.9 | 75.3 | 25.2 |
| Ettin-Enc-150m | 57.7 | 54.0 | 68.6 | 41.5 | 80.2 | 44.7 | 45.7 | 74.9 | 22.6 |
| **Large Models (305-395M parameters)** | | | | | | | | | |
| BERT-large | 57.2 | 53.8 | 68.3 | 41.5 | 81.1 | 44.3 | 42.9 | 76.1 | 22.5 |
| ModernBERT-large | 58.6 | 55.0 | 69.1 | 41.5 | 82.2 | 45.5 | 47.0 | 76.5 | 23.5 |
| Ettin-Enc-400m | 59.4 | 55.5 | 69.9 | 41.8 | 82.6 | 45.6 | 48.4 | 77.2 | 22.6 |
| **XL Models (750-1565M parameters)** | | | | | | | | | |
| DeBa-v1-xl | 59.5 | 56.4 | 70.8 | 42.5 | 82.7 | 45.7 | 47.2 | 77.1 | 28.6 |
| DeBa-v2-xxl* | 60.5 | 57.4 | 71.7 | 44.4 | 82.4 | 46.5 | 47.7 | 78.3 | 30.9 |
| Ettin-Enc-1b | 60.4 | 56.0 | 72.2 | 41.9 | 83.3 | 46.4 | 50.1 | 77.7 | 20.4 |

Table 6: MTEB v2 English results. Class. = Classification, Clus. = Clustering, Pair. Class. = Pair Classification, Rerank. = Reranking, STS = Semantic Textual Similarity, Summ. = Summarization. DeBERTa v2 XXL did not converge with the standard learning rate sweeps, so we used a lower learning rate in order to help it to converge.

| Model Name | Single Sentence | | Paraphrase and Similarity | | | Natural Language Inference | | | |
|---|---|---|---|---|---|---|---|---|---|
| | CoLA | SST-2 | MRPC | STS-B | QQP | MNLI | QNLI | RTE | Avg |
| **XXS Models (11-25M parameters)** | | | | | | | | | |
| BERT-mini | 33.9 | 88.3 | 83.8 | 86.4 | 89.3 | 77.2 | 85.4 | 67.1 | 76.4 |
| TinyBERT | 22.4 | 91.2 | 87.5 | 87.8 | 89.4 | 80.9 | 88.4 | 68.2 | 77.0 |
| Ettin-Enc-17m | 43.9 | 91.2 | 86.0 | 87.2 | 89.8 | 79.5 | 87.3 | 69.0 | 79.2 |
| **XS Models (28-33M parameters)** | | | | | | | | | |
| BERT-small | 44.8 | 90.1 | 83.1 | 87.6 | 90.1 | 79.2 | 88.2 | 69.3 | 79.0 |
| MiniLM L12 | 59.1 | 93.3 | 91.2 | 89.2 | 91.5 | 85.6 | 91.9 | 74.7 | 84.6 |
| Ettin-Enc-32m | 57.4 | 92.0 | 89.7 | 89.5 | 91.0 | 83.4 | 90.7 | 74.7 | 83.5 |
| **S Models (68-82M parameters)** | | | | | | | | | |
| DistilBERT | 56.9 | 92.2 | 86.8 | 87.4 | 90.8 | 82.7 | 89.5 | 66.1 | 81.5 |
| DistilRoBERTa | 61.9 | 93.1 | 89.0 | 88.9 | 91.5 | 84.7 | 91.7 | 69.7 | 83.8 |
| Ettin-Enc-68m | 64.8 | 94.4 | 92.2 | 91.1 | 91.9 | 87.0 | 92.9 | 83.8 | 87.2 |
| **Base Models (86-150M parameters)** | | | | | | | | | |
| BERT-base | 59.0 | 93.1 | 89.5 | 89.4 | 91.4 | 85.4 | 91.6 | 78.2 | 84.7 |
| ModernBERT-base | 65.1 | 96.0 | 92.2 | 91.8 | 92.1 | 89.1 | 93.9 | 87.4 | 88.4 |
| Ettin-Enc-150m | 66.9 | 95.8 | 92.6 | 92.2 | 92.4 | 89.2 | 94.0 | 87.7 | 88.9 |
| **Large Models (305-395M parameters)** | | | | | | | | | |
| BERT-large | 56.2 | 93.3 | 87.8 | 90.6 | 90.9 | 86.3 | 92.8 | 83.8 | 85.2 |
| ModernBERT-large | 71.4 | 97.1 | 91.7 | 92.8 | 92.7 | 90.8 | 95.2 | 92.1 | 90.4 |
| Ettin-Enc-400m | 71.3 | 96.7 | 93.6 | 92.7 | 93.0 | 91.3 | 95.2 | 92.8 | 90.8 |
| **XL Models (750-1565M parameters)** | | | | | | | | | |
| DeBERTa-v2-XL | 75.3 | 97.1 | 91.7 | 92.5 | 92.6 | 91.7 | 95.9 | 89.2 | 90.7 |
| DeBERTa-v2-XXL | 71.6 | - | - | - | - | 91.2 | 96.0 | - | - |
| Ettin-Enc-1b | 74.4 | 97.1 | 94.4 | 93.2 | 93.0 | 91.8 | 96.0 | 93.1 | 91.6 |

Table 7: GLUE benchmark results across model sizes and architectures. DeBERTa v2 XXL was run for 300+ GPU hours before ruling it out due to it's large size (> 1.5B), results are incomplete.

| Model Name | ARC | HS | LMB | OBQA | PIQA | SIQA | SciQ | TQA | WG | WSC | Avg |
|---|---|---|---|---|---|---|---|---|---|---|---|
| **XXS Models (17M parameters)** | | | | | | | | | | | |
| Ettin-Enc-from-Dec-17m | 27.7 | 27.2 | 23.4 | 31.4 | 56.0 | 34.6 | 45.9 | 0.5 | 51.2 | 52.7 | 35.1 |
| Ettin-Enc-17m | 28.3 | 26.4 | 24.1 | 34.0 | 54.2 | 34.4 | 44.0 | 0.1 | 52.6 | 52.7 | 35.1 |
| Ettin-Dec-From-Enc-17m | 22.7 | 26.8 | 21.9 | 24.6 | 56.1 | 70.9 | 35.7 | 0.9 | 53.4 | 53.8 | 36.7 |
| Ettin-Dec-17m | 21.3 | 27.1 | 23.0 | 27.2 | 57.7 | 71.1 | 35.4 | 2.6 | 50.9 | 48.0 | 36.4 |
| **XS Models (32M parameters)** | | | | | | | | | | | |
| Ettin-Enc-from-Dec-32m | 28.2 | 27.9 | 29.5 | 33.8 | 55.6 | 34.7 | 45.6 | 0.1 | 53.2 | 50.9 | 36.0 |
| Ettin-Enc-32m | 28.7 | 28.0 | 33.6 | 34.8 | 56.7 | 34.4 | 41.4 | 0.2 | 51.4 | 56.4 | 36.6 |
| Ettin-Dec-From-Enc-32M | 20.5 | 28.3 | 27.7 | 27.0 | 58.1 | 77.2 | 36.0 | 3.0 | 50.2 | 52.7 | 38.1 |
| Ettin-Dec-32m | 23.5 | 28.5 | 28.5 | 28.2 | 57.7 | 77.5 | 36.4 | 3.8 | 53.1 | 50.2 | 38.7 |
| **Small Models (68M parameters)** | | | | | | | | | | | |
| Ettin-Enc-from-Dec-68m | 30.0 | 30.4 | 31.8 | 33.6 | 57.1 | 36.1 | 55.3 | 1.9 | 51.1 | 52.7 | 38.0 |
| Ettin-Enc-68m | 29.5 | 31.6 | 36.1 | 35.4 | 58.4 | 35.6 | 49.6 | 1.1 | 51.3 | 62.6 | 39.1 |
| Ettin-Dec-from-Enc-68m | 24.8 | 31.9 | 35.8 | 29.4 | 60.7 | 84.6 | 38.3 | 5.8 | 53.1 | 56.0 | 42.1 |
| Ettin-Dec-68m | 25.3 | 33.4 | 35.2 | 29.4 | 61.8 | 83.2 | 38.8 | 5.6 | 50.1 | 55.3 | 41.8 |
| **Base Models (150M parameters)** | | | | | | | | | | | |
| Ettin-Enc-from-Dec-150m | 33.5 | 36.3 | 39.2 | 34.4 | 63.9 | 39.7 | 74.3 | 4.7 | 51.5 | 59.3 | 43.7 |
| Ettin-Enc-150m | 32.5 | 36.5 | 41.6 | 37.4 | 63.0 | 38.5 | 59.8 | 1.6 | 54.9 | 63.0 | 42.9 |
| Ettin-Dec-from-Enc-150m | 25.0 | 36.0 | 39.4 | 30.0 | 62.9 | 84.7 | 40.4 | 7.4 | 52.9 | 57.5 | 43.6 |
| Ettin-Dec-150m | 28.6 | 40.3 | 43.2 | 29.2 | 66.6 | 89.6 | 40.1 | 11.2 | 53.7 | 59.0 | 46.2 |
| **Large Models (400M parameters)** | | | | | | | | | | | |
| Ettin-Enc-from-Dec-400m | 39.7 | 47.7 | 44.9 | 38.6 | 66.3 | 43.4 | 70.4 | 7.7 | 56.4 | 68.9 | 48.4 |
| Ettin-Enc-400m | 35.6 | 46.8 | 50.5 | 38.0 | 64.7 | 43.9 | 65.6 | 6.4 | 59.7 | 70.7 | 48.2 |
| Ettin–Dec-from-Enc-400m | 29.9 | 45.8 | 46.4 | 33.6 | 66.9 | 92.1 | 45.3 | 13.3 | 53.9 | 63.7 | 49.1 |
| Ettin-Dec-400m | 33.6 | 54.3 | 52.3 | 34.4 | 71.0 | 91.8 | 45.5 | 18.3 | 57.6 | 71.8 | 53.1 |
| **XL Models (1B parameters)** | | | | | | | | | | | |
| Ettin-Enc-from-Dec-1B | 42.4 | 53.0 | 49.3 | 39.2 | 70.0 | 46.3 | 74.9 | 14.9 | 62.3 | 73.3 | 52.5 |
| Ettin-Enc-1B | 37.3 | 52.3 | 54.0 | 38.4 | 67.6 | 46.3 | 64.5 | 7.6 | 63.2 | 75.8 | 50.7 |
| Ettin-Dec-from-Enc-1B | 32.5 | 52.5 | 49.1 | 35.8 | 69.9 | 93.1 | 48.5 | 13.1 | 58.6 | 69.2 | 52.2 |
| Ettin-Dec-1B | 39.7 | 62.9 | 58.4 | 41.6 | 74.4 | 93.8 | 48.2 | 29.3 | 62.7 | 79.1 | 59.0 |

Table 8: Performance comparison of all models evaluated on generative tasks. Enc-from-Dec are trained with MTNP from decoders, while Dec-from-Enc are encoders trained with CLM.

| Model Name | Retrieval (nDCG@10) | MNLI (Accuracy) | Generative Avg |
|---|---|---|---|
| **XXS Models (17M parameters)** | | | |
| Ettin-Enc-17m | 30.93 | 79.5 | 35.1 |
| Ettin-Dec-17m | 29.11 | 77.6 | 36.4 |
| Ettin-Enc-from-Dec-17m | 31.01 | 77.7 | 35.1 |
| Ettin-Dec-from-Enc-17m | 28.52 | 78.8 | 36.7 |
| **XS Models (32M parameters)** | | | |
| Ettin-Enc-32m | 35.13 | 83.4 | 36.6 |
| Ettin-Dec-32m | 32.93 | 80.4 | 38.7 |
| Ettin-Enc-from-Dec-32m | 34.66 | 80.9 | 36.0 |
| Ettin-Dec-from-Enc-32m | 32.32 | 82.6 | 38.1 |
| **Small Models (66-70M parameters)** | | | |
| Ettin-Enc-68m | 38.17 | 87.0 | 39.1 |
| Ettin-Dec-68m | 36.12 | 83.9 | 41.8 |
| Ettin-Enc-from-Dec-68m | 37.87 | 83.9 | 38.0 |
| Ettin-Dec-from-Enc-68m | 36.31 | 85.8 | 42.1 |
| **Medium Models (150M parameters)** | | | |
| Ettin-Enc-150m | 39.97 | 89.2 | 42.9 |
| Ettin-Dec-150m | 37.71 | 85.6 | 46.2 |
| Ettin-Enc-from-Dec-150m | 39.49 | 85.8 | 43.7 |
| Ettin-Dec-from-Enc-150m | 37.55 | 86.8 | 43.6 |
| **Large Models (400M parameters)** | | | |
| Ettin-Enc-400m | 42.24 | 91.3 | 48.2 |
| Ettin-Dec-400m | 39.93 | 88.2 | 53.1 |
| Ettin-Enc-from-Dec-400m | 41.44 | 87.6 | 48.4 |
| Ettin-Dec-from-Enc-400m | 39.69 | 89.4 | 49.1 |
| **XL Models (1B parameters)** | | | |
| Ettin-Enc-1b | 43.35 | 91.8 | 50.7 |
| Ettin-Dec-1b | 41.70 | 89.9 | 59.0 |
| Ettin-Enc-from-Dec-1b | 43.24 | 89.0 | 52.5 |
| Ettin-Dec-from-Enc-1b | 40.77 | 90.5 | 52.2 |

Table 9: Table version of Figure 1. The generative eval breakdowns can be found in Table 8.

| Model Name | WinoGender All | | | | WinoGender Gotcha | | |
|---|---|---|---|---|---|---|---|
| | Overall | Female | Male | Neutral | Overall | Female | Male |
| **XXS Models (17M parameters)** | | | | | | | |
| Ettin-Enc-from-Dec-17m | 50.8 ± 1.9 | 51.7 ± 3.2 | 50.8 ± 3.2 | 50.0 ± 3.2 | 50.4 ± 3.2 | 45.8 ± 4.6 | 55.0 ± 4.6 |
| Ettin-Enc-17m | 50.6 ± 1.9 | 50.0 ± 3.2 | 50.8 ± 3.2 | 50.8 ± 3.2 | 50.0 ± 3.2 | 46.7 ± 4.6 | 53.3 ± 4.6 |
| Ettin-Dec-from-Enc-17m | 49.9 ± 1.9 | 50.0 ± 3.2 | 50.0 ± 3.2 | 49.6 ± 3.2 | 49.6 ± 3.2 | 47.5 ± 4.6 | 51.7 ± 4.6 |
| Ettin-Dec-17m | 51.1 ± 1.9 | 50.0 ± 3.2 | 51.2 ± 3.2 | 52.1 ± 3.2 | 49.2 ± 3.2 | 45.0 ± 4.6 | 53.3 ± 4.6 |
| **XS Models (32M parameters)** | | | | | | | |
| Ettin-Enc-from-Dec-32m | 53.6 ± 1.9 | 52.5 ± 3.2 | 53.8 ± 3.2 | 54.6 ± 3.2 | 53.3 ± 3.2 | 50.0 ± 4.6 | 56.7 ± 4.5 |
| Ettin-Enc-32m | 53.2 ± 1.9 | 53.8 ± 3.2 | 52.9 ± 3.2 | 52.9 ± 3.2 | 52.9 ± 3.2 | 52.5 ± 4.6 | 53.3 ± 4.6 |
| Ettin-Dec-from-Enc-32m | 51.1 ± 1.9 | 51.7 ± 3.2 | 51.2 ± 3.2 | 50.4 ± 3.2 | 51.7 ± 3.2 | 49.2 ± 4.6 | 54.2 ± 4.6 |
| Ettin-Dec-32m | 50.8 ± 1.9 | 50.4 ± 3.2 | 50.8 ± 3.2 | 51.2 ± 3.2 | 50.0 ± 3.2 | 50.0 ± 4.6 | 50.0 ± 4.6 |
| **Small Models (66-70M parameters)** | | | | | | | |
| Ettin-Enc-from-Dec-68m | 51.9 ± 1.9 | 52.5 ± 3.2 | 51.7 ± 3.2 | 51.7 ± 3.2 | 51.2 ± 3.2 | 53.3 ± 4.6 | 49.2 ± 4.6 |
| Ettin-Enc-68m | 56.1 ± 1.9 | 55.8 ± 3.2 | 56.7 ± 3.2 | 55.8 ± 3.2 | 56.7 ± 3.2 | 56.7 ± 4.5 | 56.7 ± 4.5 |
| Ettin-Dec-from-Enc-68m | 51.8 ± 1.9 | 51.7 ± 3.2 | 52.1 ± 3.2 | 51.7 ± 3.2 | 50.8 ± 3.2 | 51.7 ± 4.6 | 50.0 ± 4.6 |
| Ettin-Dec-68m | 54.2 ± 1.9 | 55.0 ± 3.2 | 53.8 ± 3.2 | 53.8 ± 3.2 | 52.9 ± 3.2 | 56.7 ± 4.5 | 49.2 ± 4.6 |
| **Medium Models (150M parameters)** | | | | | | | |
| Ettin-Enc-from-Dec-150m | 52.8 ± 1.9 | 50.8 ± 3.2 | 54.2 ± 3.2 | 53.3 ± 3.2 | 52.1 ± 3.2 | 47.5 ± 4.6 | 56.7 ± 4.5 |
| Ettin-Enc-150m | 57.5 ± 1.8 | 57.1 ± 3.2 | 57.9 ± 3.2 | 57.5 ± 3.2 | 57.5 ± 3.2 | 55.8 ± 4.6 | 59.2 ± 4.5 |
| Ettin–Dec-from-Enc-150m | 53.3 ± 1.9 | 52.9 ± 3.2 | 52.9 ± 3.2 | 54.2 ± 3.2 | 52.1 ± 3.2 | 55.8 ± 4.6 | 48.3 ± 4.6 |
| Ettin-Dec-150m | 54.7 ± 1.9 | 53.3 ± 3.2 | 55.8 ± 3.2 | 55.0 ± 3.2 | 52.9 ± 3.2 | 56.7 ± 4.5 | 49.2 ± 4.6 |
| **Large Models (400M parameters)** | | | | | | | |
| Ettin-Enc-from-Dec-400m | 55.1 ± 1.9 | 55.4 ± 3.2 | 54.6 ± 3.2 | 55.4 ± 3.2 | 55.4 ± 3.2 | 53.3 ± 4.6 | 57.5 ± 4.5 |
| Ettin-Enc-400m | 70.3 ± 1.7 | 68.8 ± 3.0 | 70.8 ± 2.9 | 71.2 ± 2.9 | 69.2 ± 3.0 | 69.2 ± 4.2 | 69.2 ± 4.2 |
| Ettin-Dec-from-Enc-400m | 54.0 ± 1.9 | 53.8 ± 3.2 | 55.0 ± 3.2 | 53.3 ± 3.2 | 52.5 ± 3.2 | 55.8 ± 4.6 | 49.2 ± 4.6 |
| Ettin-Dec-400m | 55.3 ± 1.9 | 54.6 ± 3.2 | 55.8 ± 3.2 | 55.4 ± 3.2 | 52.9 ± 3.2 | 51.7 ± 4.6 | 54.2 ± 4.6 |
| **XL Models (1B parameters)** | | | | | | | |
| Ettin-Enc-from-Dec-1B | 57.9 ± 1.8 | 56.7 ± 3.2 | 58.3 ± 3.2 | 58.8 ± 3.2 | 54.2 ± 3.2 | 48.3 ± 4.6 | 60.0 ± 4.5 |
| Ettin-Enc-1B | 68.2 ± 1.7 | 67.1 ± 3.0 | 66.2 ± 3.1 | 71.2 ± 2.9 | 65.8 ± 3.1 | 65.8 ± 4.3 | 65.8 ± 4.3 |
| Ettin-Dec-from-Enc-1B | 55.8 ± 1.9 | 55.8 ± 3.2 | 56.7 ± 3.2 | 55.0 ± 3.2 | 54.2 ± 3.2 | 54.2 ± 4.6 | 54.2 ± 4.6 |
| Ettin-Dec-1B | 56.7 ± 1.8 | 56.7 ± 3.2 | 55.4 ± 3.2 | 57.9 ± 3.2 | 52.1 ± 3.2 | 50.8 ± 4.6 | 53.3 ± 4.6 |

Table 10: WinoGender accuracy results (values: Accuracy % ± Std Error %). Results taken from the Eleuther AI harness. Many of the small models do not get above random performance (50%).

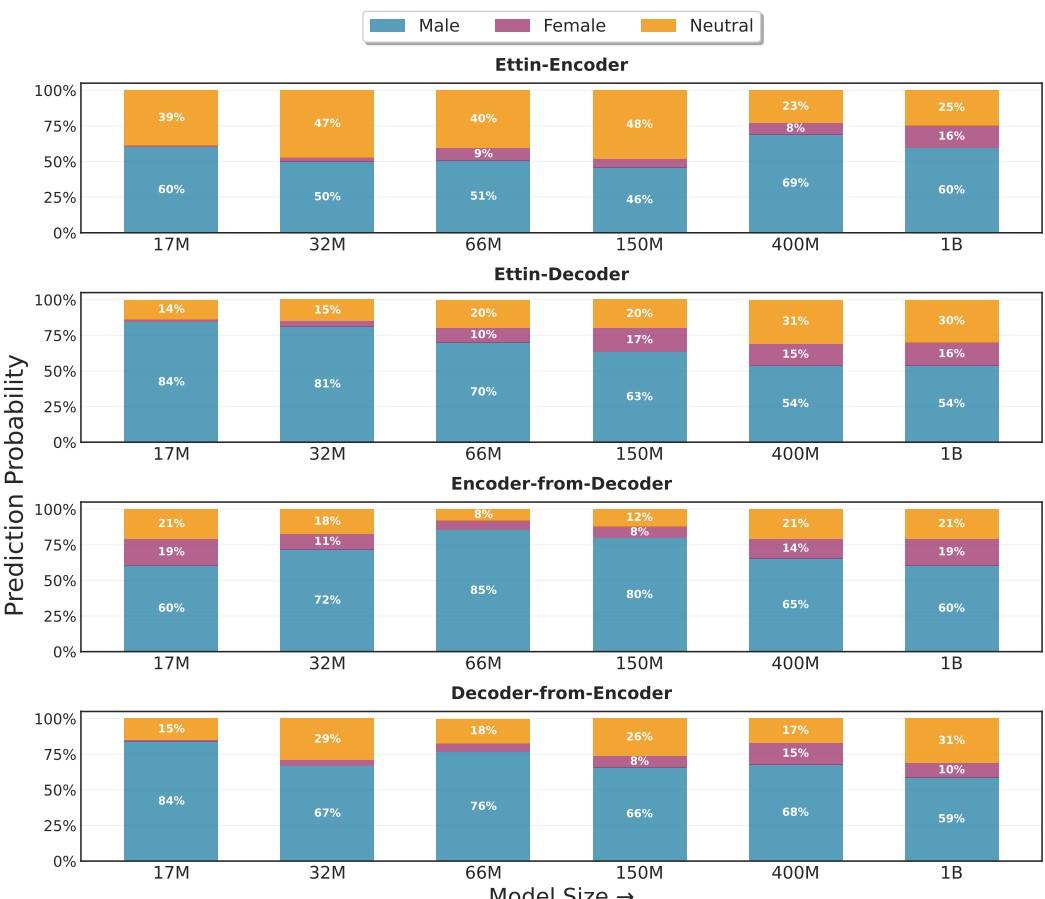

Figure 3: Full gender pronoun predictions results on the Gotcha split of WinoGender (Rudinger et al., 2018), a 50/50 stereotypical split. **We see that encoder models are more likely to use gender neutral pronouns whereas both are biased towards male pronouns.**

| Model Name | Total Params | Embed Params | Non-Embed Params |
|---|---|---|---|
| **XXS Models (7-17M parameters)** | | | |
| Pythia-14m | 7.6M | 6.4M | 1.2M |
| BERT Tiny | 11.2M | 7.9M | 3.2M |
| TinyBERT | 14.4M | 9.7M | 4.7M |
| Ettin-17m | 16.8M | 12.9M | 3.9M |
| **XS Models (28-33M parameters)** | | | |
| BERT Small | 28.8M | 15.9M | 12.9M |
| Ettin-32m | 31.9M | 19.3M | 12.5M |
| MiniLM L12 | 33.4M | 11.9M | 21.4M |
| **Small Models (68-82M parameters)** | | | |
| DistilBERT Base | 66.4M | 23.8M | 42.5M |
| Ettin-68m | 68.1M | 25.8M | 42.4M |
| DistilGPT2 | 81.9M | 39.4M | 42.5M |
| DistilRoBERTa Base | 82.1M | 39.0M | 43.1M |
| **Base Models (123-150M parameters)** | | | |
| BERT-base | 109.5M | 23.8M | 85.6M |
| Pythia-160m | 123.7M | 38.6M | 85.1M |
| SmolLM2-135m | 134.5M | 28.3M | 106.2M |
| ModernBERT-base | 149.0M | 38.7M | 110.3M |
| Ettin-150m | 149.0M | 38.7M | 110.3M |
| **Large Models (353-395M parameters)** | | | |
| BERT-large | 335.1M | 31.8M | 303.4M |
| Pythia-410m | 353.8M | 51.5M | 302.3M |
| SmolLM2-360m | 361.8M | 47.2M | 314.6M |
| ModernBERT-large | 394.8M | 51.6M | 343.2M |
| Ettin-400m | 394.8M | 51.6M | 343.2M |
| **XL Models (884M-1.2B parameters)** | | | |
| DeBERTa v2 XLarge | 884.6M | 197.6M | 687.0M |
| Pythia 1B | 908.8M | 103.0M | 805.7M |
| Ettin-1B | 1028.1M | 90.3M | 937.8M |
| OLMo 1B 0724 | 1176.8M | 103.0M | 1073.7M |
| Llama 3.2 1B | 1235.8M | 262.7M | 973.1M |

Table 11: Parameter breakdown of language models organized by size categories. Models are grouped by total parameter count and show the distribution between embedding and non-embedding parameters across different architectures. Parameter counts are the same for Ettin encoders, decoders, and cross-objective trained versions.

| Parameter | Value |
|---|---|
| Vocabulary Size | 50,368 |
| Max Sequence Length | 1024->7999 |
| Tokenizer | ModernBERT |
| Attention Layer | RoPE |
| Attention Dropout | 0.0 |
| Attention Output Bias | false |
| Attention Output Dropout | 0.1 |
| Attention QKV Bias | false |
| Transformer Layer | prenorm |
| Embedding Dropout | 0.0 |
| Embedding Norm | true |
| Final Norm | true |
| Skip First PreNorm | true |
| Embedding Layer | sans_pos |
| MLP Dropout | 0.0 |
| MLP Input Bias | false |
| MLP Layer Type | GLU |
| MLP Output Bias | false |
| Normalization | LayerNorm |
| Norm Epsilon | 1e-12 |
| Norm Bias | false |
| Hidden Activation | GELU |
| Head Pred Activation | GELU |
| Activation Function | GELU |
| Padding | unpadded |
| Rotary Embedding Base | 160,000.0 |
| Rotary Embedding Interleaved | false |
| Allow Embedding Resizing | true |
| Sliding Window | 128 |
| Global Attention Every N Layers | 3 |
| Unpad Embeddings | true |

Table 12: Common pre-training configuration parameters across all models

