# OpenReview forum: "Seq vs Seq: An Open Suite of Paired Encoders and Decoders"
_ICLR.cc/2026/Conference — ICLR 2026 Poster_

### Official Review · Reviewer_npLe · 2025-10-31

**Soundness:** 3
**Presentation:** 3
**Contribution:** 3
**Rating:** 6
**Confidence:** 4

**Summary:**

This paper introduces SEQ vs SEQ (ETTIN Suite), an open-source suite of paired encoder-only and decoder-only language models trained with identical data, architecture, and training recipes. The suite spans models from 17M to 1B parameters, enabling an “apples-to-apples” comparison of encoder and decoder architectures across scaling trends. The authors also provide 200+ checkpoints, token-level training data ordering, and open datasets, making this a valuable resource for community research. Like existing works, encoder-only models outperform decoders on classification and retrieval tasks (e.g., GLUE, MTEB, MS MARCO). Decoder-only models dominate generative tasks (e.g., ARC, HellaSwag, TriviaQA). Adapting models to the opposite objective through continued cross-objective pretraining (e.g., MLM-trained decoders, CLM-trained encoders) is ineffective can not beat the single native baseline. Additionally, the paper includes a case study on gender bias, showing diverging pronoun prediction behaviors between encoder and decoder models.

**Strengths:**

1. An apple-to-apple suite of encoder and decoder models trained with identical recipes and data. This eliminates confounding factors present in previous studies.
2. All model weights, checkpoints, and training data orders are released. This enables reproducibility and further analysis.
3. Both encoder and decoder models achieve state-of-the-art performance for their size, outperforming ModernBERT (encoder) and SmolLM2/LLaMA 3.2 1B (decoder) baselines.
4. Demonstrates that continued training of decoders with MLM (or encoders with CLM) fails to match native training, even when using 50B tokens of additional training.
5. Covers 6 model sizes (17M → 1B). This allows to study scaling trends across both architectures.

**Weaknesses:**

1. Cross-training adapting decoder into encoder often applies masked language modeling. Is it helpful also add masked next token prediction or unsupervised contrastive learning like [1].
2. The evaluation is centered on GLUE, MTEB, and knowledge-based benchmark. Can you also math or coding benchmarks to see the reasoning abilities of the models.

References:
BehnamGhader et al. LLM2Vec: Large Language Models Are Secretly Powerful Text Encoders. COLM 24

**Questions:**

1. Why are the results of TQA in Table 4 exceptionally low for some models?
2. For smaller models (<=70M), adapting encoder to decoder or decoder to encoder seem little gap from native pretraining. If we increase the tokens for adapting phase, would that be helpful? Is there any scaling behavior that can bridge the gap between adapting model performance and native one?

---

> ### Author Response · Authors · 2025-11-18
>
> Thank you for recognizing that our work “eliminates confounding factors”, “achieve state-of-the-art performance”, and “allows [for] stud[ying] scaling trends across both architectures”!
>
> > Cross-training adapting decoder into encoder often applies masked language modeling. Is it helpful also add masked next token prediction or unsupervised contrastive learning like [1].
>
> **There may be a misunderstanding as we do in fact use both of these (Section 3.5)!** We will update this section to make this more clear. We use MNTP instead of MLM (footnote 11) as previous work (and some of our initial results) found it more effective. We also do use contrastive learning in the retrieval training like LLM2Vec. Their work was a motivating work for ours, and we aimed to reproduce their approach like you suggest.
>
> > The evaluation is centered on GLUE, MTEB, and knowledge-based benchmark. Can you also math or coding benchmarks to see the reasoning abilities of the models.
>
> This is a great suggestion. Most of our models are very small (< 1B) and perform at near chance on these as they are not specifically trained for math or code. However, upon your suggestion, we ran the decoder model versions on additional benchmarks including math (e.g. GSM8K, generative) and computer science MMLU (multiple choice classification, random is 25%) for the 1B size:
>
> | Model        | MMLU CS | GSM8k |
> |--------------|---------|-------|
> | Decoder      | 27.0    | 32.0  |
> | Dec-From-Enc | 37.0    | 18.9  |
>
> **You can see that these results align with those found in the paper**: on classification (e.g. MMLU) the Dec-From-Enc trained with MLM does better; on generative tasks (e.g. GSM8k) generative only (i.e. Decoder) models do better. Note that our scores are also greater than those reported in SmolLM2 paper (31.1 and 7.6 for LLaMA 3.2 1B).  We will update the paper to include these results!
>
> > Why are the results of TQA in Table 4 exceptionally low for some models?
>
> Great question! We use it in a zero-shot closed-book eval (no RAG) and the scores are much lower than other frameworks which use RAG or few-shots. Nonetheless, our results are only slightly below those in the SmolLM2 paper where they use a few-shot prompt without RAG (e.g. 28.1 few-shot with Llama 3.2 1B in SmolLM2 while we report 24.9 zero-shot).
>
> > For smaller models (<=70M), adapting encoder to decoder or decoder to encoder seem little gap from native pretraining. If we increase the tokens for adapting phase, would that be helpful? Is there any scaling behavior that can bridge the gap between adapting model performance and native one?
>
> It is possible that adapting further will boost performance. However, we already continued trained for 5x what LLM2Vec proposed and it did not help, as you mention. Although out of scope for this work, our guess is that the best way to bridge this gap is to use mixed pre-training objectives, as otherwise it seems like the smaller models overfit to their objective function and have a hard time recovering in the minimal weight space.

---

### Official Review · Reviewer_dWKC · 2025-11-01

**Soundness:** 3
**Presentation:** 3
**Contribution:** 3
**Rating:** 6
**Confidence:** 3

**Summary:**

The paper presents a set of open-source and open-data text models, consisting of encoders (trained with a masked language modeling objective) and decoders (trained as autoregressive language models) of sizes from 17M to 1B parameters.  The main goal is to enable direct comparison of encoders and decoders on a variety of tasks:  classification and retrieval tasks, where encoders tend to be preferred, and generation tasks where causal LMs are typically best.  For each model size the authors train a pair of encoder and decoder trained with the same hyperparameters, as well as an encoder further trained with a causal LM loss and a decoder further trained with MLM.  Whereas previous such comparisons have compared models of varying sizes and training setups, this work allows for more direct comparison.  The main findings are that (1) as expected, encoders outperform decoders of the same size on classification and retrieval tasks and vice versa for generation tasks and (2) continued pre-training with the other loss (e.g. decoders further trained with MLM) does not undo this trend.  As a case study of additional analyses that this model family enables, the authors also study the distribution of gender pronouns for models of different type and size, finding for example that encoders tend to favor more neutral pronouns and that the gender distribution changes with model size.

The main contribution of the paper is the open models themselves.  The findings are interesting and demonstrate the value of the resource, but are not very far-reaching.  I therefore consider this paper to be mainly a "resource" paper.  A natural question is whether the resource rises to the level of a research contribution and therefore is worthy of a conference publication.  I believe the contribution is important and am recommending acceptance.  That being said, the paper could make a more convincing research contribution if it provided more detail and analysis, beyond just the models themselves (see below under "weaknesses"), and for this reason my recommendation is marginal acceptance.

**Strengths:**

- A large set of open models trained on open data makes it possible to make comparisons that were previously only approximate (e.g. comparing encoders an decoders of different sizes, or of similar sizes but with otherwise very different training conditions).

- The findings are interesting.  It is especially interesting to see that decoders can't be easily "converted" into equally strong encoders via continued pre-training, and vice versa.

- The paper is generally presented well.  It provides good background on existing models and findings and lays out the new work clearly.

**Weaknesses:**

- The paper should provide more detail about how the hyperparameters were chosen.  For example, it is not obvious to me that using identical hyperparameters for encoders and decoders is the best choice.  Ideally there would be some tuning done and a study of the sensitivity to hyperparameter choices.  Could the results be improved with different choices of hyperparameters?  Could they change enough to modify the findings?

- The results are given without any analysis into why the findings are as they are, especially the less obvious ones related to continued pre-training and gender pronouns.  I recognize that it may not be easy to do, but I would expect some attempt at understanding the results and studying how they may be affected by aspects of the data distribution, number of training iterations, or other variables.  This is related to the first point above, but has more to do with explanation of the findings.

- A minor weakness:  While the paper is generally well-written, there are quite a few minor grammatical errors throughout.  A thorough proofreading would help.

- Also minor:  Citations should be provided to support the statement "Part of this gap is due to the sentiment within the community that decoders can be adapted for use in tasks that were once predominantly encoder-focused (e.g. classification, embeddings)".

**Questions:**

- Did you consider including larger models, say up to 3B or 7B?  The work is valuable without it, and to some extent the maximum size is arbitrary.  However, several B parameters is a fairly typical size for research models, and it would be interesting to know whether there is any "phase transition" as model size increases further.

---

> ### Author Response · Authors · 2025-11-18
>
> Thank you for recognizing our work is "presented well" and "makes it possible to make comparisons that were previously only approximate"!
>
> > The paper should provide more detail about how the hyperparameters were chosen. For example, it is not obvious to me that using identical hyperparameters for encoders and decoders is the best choice.
>
> We believe we do use different hyperparameters when it matters (e.g. mask rate). However, for standard LLM hyperparameters like LR, dropout, etc. we do not believe there is a large difference for comparing the two: both started with exactly the same weight matrices, with exactly the same layers, for identical models. The only difference in training is the bidirectional attention and the objective function: otherwise they are both deep NN that have the same need for warmup, LR, etc. This was reinforced by the ModernBERT paper, which simply applied decoder-only tricks to encoders and got massive gains: these models are optimized in the same way.
>
> Although it is possible that more optimization could have resulted in slightly better results, **all of our models already reach state-of-the-art** results: our encoder is SoTA in English (beating even the recent ModernBERT) and our decoders are SoTA among open-data models for their size (even beating models like SmolLM2 and Llama 3.2 created by large industry groups). Given that both are SoTA, we believe this strengthens the claim that they don’t need different hyperparameters.
>
> > Did you consider including larger models, say up to 3B or 7B?
>
> We agree that 7B models would show even higher scores! However, there are two main reasons we did not:
>
> (1) Practically, people are not using 7B encoder models: XLM-R XL [1] has a 3B and 10B version yet most people don’t know it exists. If you check download stats, it is by far the least used model. We also note that smaller encoders are more “batch friendly” for high throughput tasks like indexing a large corpus. Thus, our approach already compares encoders and decoders for the most frequently used encoder sizes and already shows a broad range of scaling.
>
> (2) The other main issue is that our analysis requires not just one but two models to be trained from scratch for each size, at a scale that allows for SoTA performance. At the 7B scale this is very expensive and we do not have access to large-scale compute, only a single node of 4xH100s (see Appendix E for details). Given that people do not currently use > 1B encoders frequently and we do not have the compute for two SoTA 7B models, we did not pursue it.
>
> However, we would love to see the community (or an industry lab) create such a model as our experiments show it would be better than 7B decoders (and even comparable to 14B decoders on tasks like classification), which is also why we opened up everything, including code and data, to easily allow such experimentations.
>
> Thus, although it would be nice to have 7B models we don’t feel it is required for our contribution given current encoder usage.
>
> > The results are given without any analysis into why the findings are as they are, especially the less obvious ones
>
> We apologize for not including much further analysis - as you say it is “not be easy to do"! We hope that providing open models and datasets will help others explore these questions further. Concretely, we don't have hard evidence on why MLM helps reduce gender bias. We have some hypotheses along the lines of forward (bidirectional) context in MLMs allowing for more detailed relationships (i.e. text cues later in the document) but didn’t want to include unsubstantiated speculation in the official paper.
>
> > A minor weakness:
>
> Thank you for the suggestions on spelling and grammar. We will update them.
>
> ### References
>
> [1] Larger-Scale Transformers for Multilingual Masked Language Modeling by Goyal et. al. 2021

---

> > ### Comment · Reviewer_dWKC · 2025-11-27
> > **Response to rebuttal**
> >
> > Thank you for the thoughtful replies.  I understand the reasoning behind fixing the hyperparameters, not including larger models, and leaving out further analysis of the findings.  Regarding hyperparameters, I don't fully agree with the reasoning for keeping them fixed for decoders and encoders, but I agree that considering the SOTA results, it is not a major concern.  It would still be good to state how the chosen values were chosen.
> >
> > As of now I am keeping my rating the same.  I would rather see the paper accepted than not, but still consider the contribution to be borderline without additional takeaways.

---

### Official Review · Reviewer_Mrcc · 2025-11-03

**Soundness:** 2
**Presentation:** 3
**Contribution:** 3
**Rating:** 4
**Confidence:** 4

**Summary:**

Authors opensource pairs of encoder-only and decoder-only models along with training pipeline. They explore and compare the performance in classification, retrieval, and generative task.

**Strengths:**

1. The shift from decoder to encoder on LLMs is still not scientifically clear. This paper present paired encoder and decoder models for people to study on and has done some initial exploration.
2. The paper is fully opensource in terms of weights and training pipeline.

**Weaknesses:**

1. There are research on turning decoder-only models to encoders like [1], which authors may need to take into account when claiming “encoder models are better at classification”. The possibility of “with proper finetuning, decoder-only model outperforms encoder-only model on non-generative tasks” cannot be ruled out by the current set of experiments. Therefore some statement could be premature/misleading.
2. Authors’ efforts in opensourcing large-scale encoder models (compared to other encoder-only models) are unprecedented. However, compared to some other alternatives to decoder-only LLMs (e.g. MAMBA, diffusion LLM), the current efforts lack a 7B-scale model, which is (in my opinion) usually considered as the smallest usable model for fair comparison with decoder-only LLMs.
[1] LLM2Vec: Large Language Models Are Secretly Powerful Text Encoders. BehnamGhader et al. COLM 2024
[2] Falcon Mamba: The First Competitive Attention-free 7B Language Model, Zuo et al. Arxiv
[3] Large Language Diffusion Models. Nie et al. NeurIPS 2025.

**Questions:**

I believe such opensource efforts should be encouraged in the community. I am willing to increase my score if W1 is properly addressed by proper experiments. And would like to hear author's arguments on not conducting experiments at a larger scale for encoder-only models.

---

> ### Author Response · Authors · 2025-11-18
>
> Thank you for noticing that our work shows light on a topic that is currently “scientifically unclear” along with ”unprecedented” open source contributions!
>
> > There are research on turning decoder-only models to encoders like [1], which authors may need to take into account when claiming “encoder models are better at classification”.
>
> **We think there may be a misunderstanding**: we totally agree with you that LLM2Vec is, to the our knowledge, the best method to date to transform a decoder into an encoder. **That is why we applied exactly this process** to create encoders-from-decoders, as explained in Section 3.6 (c.f. see Section 5, 2nd paragraph).
>
> Their work aims to find the best way to transform decoders into embedding models. They only use one encoder in their whole paper (BERT) and it does not undergo the same training process as the much larger (1B-8B) models which include multiple steps of mid-training. Despite that, BERT actually comes close to the same performance as the 1B model, despite being ⅓ of the size and having significantly worse pre-training data compared to recent decoders! This underscores our decision to compare models trained on the same data and with the same size, which shows that indeed encoder models are better for the same size/data budget.
>
> > However, compared to some other alternatives to decoder-only LLMs (e.g. MAMBA, diffusion LLM), the current efforts lack a 7B-scale model
>
> We agree that 7B models would show even higher scores! However, there are two main reasons we did not:
>
> (1) **Practically, people are not using 7B encoder models: XLM-R XL [1] has a 3B and 10B version yet most people don’t know it exists**. If you check download stats, it is by far the least used model. We also note that smaller encoders are more “batch friendly” for high throughput tasks like indexing a large corpus. Thus, our approach already compares encoders and decoders for the most frequently used encoder sizes and already shows a broad range of scaling.
>
> (2) The other main issue is that our analysis requires not just one but two models to be trained from scratch for each size, at a scale that allows for SoTA performance. At the 7B scale this is very expensive and we do not have access to large-scale compute, only a single node of 4xH100s (see Appendix E for details). Given that people do not currently use > 1B encoders frequently and we do not have the compute for two SoTA 7B models, we did not pursue it.
>
> However, we would love to see the community (or an industry lab) create such a model as our experiments show it would be better than 7B decoders (and even comparable to 14B decoders on tasks like classification), which is also why we opened up everything, including code and data, to easily allow such experimentations.
>
> Thus, although it would be nice to have 7B models we don’t feel it is required for our contribution given current encoder usage.
>
> ### References
>
> [1] Larger-Scale Transformers for Multilingual Masked Language Modeling by Goyal et. al. 2021

---

> > ### Comment · Reviewer_Mrcc · 2025-11-18
> >
> > Thanks authors for their rebuttal.
> >
> > My apologies for overlooking that you were indeed testing with LLM2Vec. And I agree with authors that they have tested encoder-only model to a reasonable scale.
> >
> > In light of these, I raise my score to 6.

---

### Meta-Review · Area_Chair_DQCR · 2026-01-05

**Summary:**

This paper introduces paired encoder-only and decoder-only models trained with similar recipes on open-data. This suite of models achieves SOTA results in some benchmarks while allowing for studying questions across the two families of encoder-only vs decoder-only.

All reviewers highlight various strengths including apple-to-apple encoder/decoder models, publicly available models, code and data, state-of-the-art results, diverse model sizes and interesting findings about continued training of decoders.

**Reviewer Concerns:**

Reviewer npLe:
- **Masked next token prediction**: The authors clarified that they are already using the MNTP loss in cross-objective training as described in footnote 11. This information should be better clarified in the paper.
- **Math or coding evaluations**: The authors provide additional evaluations on MMLU and GSM8k with analysis agreeing with the results in the paper.
- **Low results in Table 4**: The authors clarified that they used zero-shot closed-book eval with no RAG and the results are comparable with SmolLM2.

Reviewer dWKC:
- **More details about the choice of hyperparameters**: The authors note that hyperparameters are similar to prior works and although there is room for improvement with further tuning, the results are already state-of-the-art.
- **Limited analysis**: The authors argue that releasing the models and datasets is a contribution and the community may further explore these questions.
- **Grammatical errors and citations to support statements**: The authors should fix and add citations.
- **Models larger than 1B such as 3B or 7B**: The reviewer acknowledges that the work is valuable without it while it can benefit from additional results. The authors respond that people do not currently use encoders larger than 1B frequently and they do not have the compute resources to train two 7B models.

Reviewer Mrcc:
- **Comparison with methods for turning decoder-only models to encoders**: The authors note that they have already used the method referred by the reviewer (LLM2Vec). The reviewer acknowledged that they overlooked this detail and agreed to raise their score.
- **Missing 7B models**: The reviewer acknowledges that the effort in opensourcing large-scale encoder models is unprecedented but 7B-scale models are missing. The authors respond that people do not currently use encoders larger than 1B frequently and they do not have the compute resources to train two 7B models.

**Reviewer Scores:**

Initially two reviewers gave a score of 6 (marginally above acceptance threshold) while one gave a score of 4 (marginally below acceptance threshold). After the authors’ response, reviewer Mrcc also raised their score to 6 as one of their concerns was due to a misunderstanding.

The main unresolved concern from the reviewers is about the lack of results at larger scales such as 3B or 7B. Regardless, all reviewers acknowledged that the work makes a notable contribution even without these larger models. The AC notes that 1B models are right at the border where one sees meaningful results from benchmarks such as MMLU and the paper is missing such results which was provided in the rebuttal.

The authors are recommended to improve the discussion on MNTP loss. Both reviewers npLe and Mrcc missed this detail. The authors may consider moving this information from footnotes (8 and 11) to the main body. In general, there seems to be an overuse of footnotes. Other footnotes may also be considered to be incorporated into the main text. Moreover, the authors are recommended to improve the writing as suggested by reviewer dWKC.

---

### Decision · Program_Chairs · 2026-01-26

Accept (Poster)